# Exploring the outcomes of research engagement using the observation method in an online setting

Deborah A Marshall [ID],[1,2] Nitya Suryaprakash [ID],[3] Danielle C Lavallee [ID],[3,4,5] Karis L Barker [ID],[1] Gail Mackean,[1] Sandra Zelinsky [ID],[1,2,6] Tamara L McCarron [ID],[1] Maria J Santana [ID],[6,7] Paul Moayyedi [ID],[2,8] Stirling Bryan [ID] [3,4,5]

For numbered affiliations see end of article.

**Correspondence to**
Dr Deborah A Marshall; damarsha@ucalgary.ca

## ABSTRACT

**Objective** The objective of this study was to explore the outcomes of research engagement (patient engagement, PE) in the context of qualitative research.

**Design** We observed engagement in two groups comprised of patients, clinicians and researchers tasked with conducting a qualitative preference exploration project in inflammatory bowel disease. One group was led by a patient research partner (PLG, partner led group) and the other by an academic researcher (RLG, researcher led group). A semistructured guide and a set of critical outcomes of research engagement were used as a framework to ground our analysis.

**Setting** The study was conducted online.

**Participants** Patient research partners (n=5), researchers (n=5) and clinicians (n=4) participated in this study.

**Main outcome measures** Transcripts of meetings, descriptive and reflective observation data of engagement during meetings and email correspondence between group members were analysed to identify the outcomes of PE.

**Results** Both projects were patient-centred, collaborative, meaningful, rigorous, adaptable, ethical, legitimate, understandable, feasible, timely and sustainable. Patient research partners (PRPs) in both groups wore dual hats as patients and researchers and influenced project decisions wearing both hats. They took on advisory and operational roles. Collaboration seemed easier in the PLG than in the RLG. The RLG PRPs spent more time than their counterparts in the PLG sharing their experience with biologics and helping their group identify a meaningful project question. A formal literature review informed the design, project materials and analysis in the RLG, while the formal review informed the project materials and analysis in the PLG. A PRP in the RLG and the PLG lead leveraged personal connections to facilitate recruitment. The outcomes of both projects were meaningful to all members of the groups.

**Conclusions** Our findings show that engagement of PRPs in research has a positive influence on the project design and delivery in the context of qualitative research in both the patient-led and researcher-led group.

## STRENGTHS AND LIMITATIONS OF THIS STUDY

⇒ We used direct observation of research engagement, which provided a more robust understanding of patient research partner roles and influence on the research.

⇒ Observation was in an online environment, and overt (group members were aware they were being directly observed in all project communications).

⇒ We created journey maps to understand governance and decision-making during all the stages of research in two groups.

⇒ We used a set of critical outcomes of research engagement as a framework to ground the work; however, it was difficult to entirely separate one outcome from the other.

⇒ Our study design was appropriate for the exploratory nature of the study; however, we were unable to ensure that both groups were equally matched in terms of experience, skills and knowledge.

## INTRODUCTION

There is a substantive body of work reporting the various ways in which patients are involved in the conduct and design of research,[1–3] and various frameworks and guidelines for supporting, evaluating and reporting patient engagement (PE) in research.[4 5] There are also studies showing the value of such engagement to the patient, such as a sense of purpose and being empowered; greater awareness of and appreciation for research; improved relationship with illness; feeling valued and gaining new skills and knowledge.[6–8] There are fewer publications on the impact and outcomes of research engagement.[6 9 10] This could be attributed to the lack of validated evaluation tools that are publicly available, informed by the literature and grounded in a theoretical or conceptual framework, inclusive of patient involvement in their development and reporting.[11–14] Some studies report hypothesised impacts instead of presenting evidence of impact.[8 15] None to our knowledge capture the impact of PE across the whole research spectrum.

We used observation methodology to obtain detailed and contextualised insights of the outcomes of research engagement throughout a health research study. This qualitative methodology has not been used extensively to study research engagement, likely due to analytical and practical challenges associated with studying a phenomenon thoroughly and at length.[16] Observational methods involve the systematic, detailed observation of behaviour and communication[17] and have been used by researchers when other methods such as interviews or surveys alone cannot fully capture the context and phenomenon under study.[18–20] Observation provides an in-depth understanding of people's actions, roles and behaviour[21 22] and identifies barriers and opportunities to more equal participation, shared decision-making, and shared understanding.[23]

In this exploratory study, our objective was to explore the outcomes of research engagement in the context of qualitative research. We observed stakeholder—especially patient research partners (PRP)—engagement in two groups. Both groups designed and conducted an exploratory qualitative preference project over a predetermined 7 month period, addressing the same research question: 'What factors or attributes are important to patients with Inflammatory Bowel Disease (IBD) in considering treatment tapering of biologics?' We used this question as the context for studying the impact of engagement since there is no standard regimen for managing adults with IBD and little evidence on patient preferences regarding treatment decisions when considering biological tapering.[24 25] Moreover, the engagement of patients in the development and design of preferences studies is recommended as good research practice.[26 27] We refer to the qualitative research conducted by the two groups as 'projects' in this study.

## METHOD

We used direct observation of two groups, a 'Patient Research Partner led Group' (PLG), led by a PRP, and an 'Academic Researcher led Group' (RLG), led by an academic researcher. Our rationale for studying two groups was to assess PE in two similar but distinctly different groups where PRPs would have sufficient opportunities to contribute and participate in the governance and decision-making across the cycle of the group work. Our intention was not to judge the leads or the groups, but to look more broadly at how PRPs engage in and influence the group project work.

We recruited PRPs (n=2), clinicians (n=2) and researchers (n=2) across Canada for each group. We identified participants through national network platforms (eg, Strategy for Patient-Oriented Research, *I*nflammation, *M*icrobiome, and *A*limentation: *G*astro-*I*ntestinal and *N*europsychiatric *E*ffects (SPOR IMAGINE) Network),[28] and study team contacts using maximum variation purposive sampling to recruit PRPs, and convenience sampling to recruit researchers and clinicians. PRPs and researchers were eligible to participate if they had basic knowledge and skills to conduct qualitative research acquired either through patient-oriented-research (POR) training, education or participating in healthcare research. Living with a chronic digestive condition such as IBD was also a requirement for PRPs. All recruited members completed a screening survey, which included select items from the Patient Centred Outcome Research Institute's Ways of Engaging- ENgagement ACtivity Tool (WE-ENACT)[29] and were then assigned to the PLG or the RLG, matching the two groups to the extent possible by their POR and qualitative research experience and training and demographics.

Due to the research taking place in 2021 during the COVID-19 pandemic and the location of group members, observation of engagement was virtual. We assigned one study staff (NS and KLB) per group, skilled in qualitative research, to observe unobtrusively, documenting all exchanges of online meetings and emails among group members. The staff received training in the four questions of observation (what to observe, how to observe, how to preserve what is observed and how to tell what was observed).[16] The staff kept notes using a semistructured guide[30] of the number of people involved in the discussions, the date of the discussion and the interactions and behaviours between group members (descriptive data). They also recorded their thoughts, biases, questions, initial interpretations of the discussions, potential themes and direct quotes that seemed significant on a word document (reflective data) (online supplemental table 1). For example, the staff documented what changes PRPs proposed that were made or not made and why or how the groups appropriately integrated group member suggestions. These notes were discussed during study team meetings to guide further data collection and generation.

All group meetings were audio recorded to verify observation notes and transcripts for their accuracy, quality and trustworthiness. The two staff listened to their group's recordings to ensure that the transcripts were verbatim and their descriptive and reflective notes captured the non-verbal cues, the predefined themes and quotes accurately. A third staff performed oversight of this work at various points in the study and resolved discrepancies. Ethical practices were followed such as assigning a unique study number on all the transcripts of meetings, emails and descriptive and reflective notes.[31–34]

Observation was overt. Group members were informed in the consent and at the first group meeting that they were being observed and all data would be anonymised prior to the analysis. After the first meeting where staff introduced themselves, they faded into the background so members could act naturally while discussing the project. We believe these strategies helped put them at ease and not alter their behaviour consciously.[31]

We used Dillon's *C*ritical *O*utcomes of *R*esearch *E*ngagement (CORE) and measures as a grounding framework to assess engagement in the two groups.[35] The 11 potential outcomes and related measures were suitable for

| Step 1 | Step 2 | Step 3 | Step 4 |
|---|---|---|---|
| **Prepare the data** | **Code the data** | **Create journey maps** | **Compare the groups** |
| • Transcribe the recorded meeting<br>• Anonymize the transcript, notes, and emails<br>• Import transcripts, notes, and emails into NVivo | • Code all data by critical outcomes and research stages<br>• Modify the description of the codes iteratively<br>• Recode | • For each group type (PRPs, researchers, and clinicians) | • Compare the journey maps of PLG stakeholders with the journey maps of RLG stakeholders to identify the critical outcomes of PRP engagement in research |

**Figure 1** Observation data analysis steps.

our study, covering a broad spectrum of the research design, approach and short and long-term outcomes of engagement.

The data (transcripts of meetings, descriptive and reflective notes) from both groups were analysed thematically in four steps using NVivo V.12 software:[36] (1) prepared and organised the data for analysis; (2) coded the data by critical outcomes, research stages and critical activities; (3) created a journey map[37 38] for each group member by 'member types' (PRPs, researchers and clinicians) to understand how each member type influenced and impacted the project and (4) compared the journey maps of all stakeholders especially the PRPs to identify the critical outcomes of PRP engagement in research (figures 1 and 2)

Data collection and analysis proceeded simultaneously using the CORE as a priori framework. Two study staff (NS and KLB) coded their group data independently. A third staff (GM) coded some data from both groups at different stages of the project, merged their coding with NS or KLB, discussed discrepancies and reached an agreement on the codes, sub codes and their descriptions. Updated versions of the coding frame were shared between the two staff via the third staff and the data were recoded. After data collection was complete, the two staff created journey maps by stakeholder type for their respective groups. The staff reviewed the journey maps of both groups, and revisited the coding done to ensure that both agreed on the final journey maps. The journey maps of the patient-led group were compared with the research-led group maps to finalise the list of outcomes of research engagement. We held a virtual meeting with

each group separately as a 'member check-in exercise' to verify their results.

NS and KLB reflected on their personal values during the data collection and analysis process to identify any biases that may have affected the research, such as attachment bias to group members. We used this approach to facilitate good practice in coding and enhance the credibility of the analysis.[39 40]

We provided the two group leads training about patient preference studies, qualitative research and about the project group work and deliverables. All this information was made available for use by other members of the two groups.

### Patient and public involvement

Our study team included one PRP (SZ) living with Crohn's disease who has extensive experience and training in conducting POR on multidisciplinary research teams. She is the Lead Patient Research Partner for the Alberta SPOR SUPPORT Unit and is a graduate of the Patient and Community Engagement Research (PaCER) programme.[41–43] She was involved in the development of the research question and study design; finalising the study approach and outcome measures; recruiting PLG and RLG group members; reviewing and providing feedback on the analysed data and reviewing this manuscript critically. We held an online meeting to discuss the results and outcomes of PE with members of both groups. The group PRPs were also involved in all the stages and critical tasks of their respective qualitative projects.

 

| Research Stage | Academic Researcher led Group | Patient Research Partner led Group |
|---|---|---|
| Getting to know your team | PRPs introduce themselves to the group. | **Members know each other & their position in the group.** |
| Deciding on how to work together | One PRP proposes having roles and tasks assigned ahead. The group discusses this strategy but does not formalize roles. Both PRPs **volunteer to tasks** during the different project stages. | PRPs propose additional strategies to communicate and agree with the final plan. Both accept the **roles assigned** to them by the lead. |
| Helping the study team understand what information is relevant to patients | PRPs share their **lived experiences especially with biologics.** One PRP has side effects and wants to stop taking biologics. | PRPs share their lived experiences. Discussed their experience, **not specifically with the treatment.** |
| Refining the study question | *PRPs question the definition of tapering and are not comfortable using the word when it was not an option for patients.* **They recommend finalizing the definition of tapering before moving on to next steps.** They look at ways tapering is defined in the literature, discuss, *and agree with the final question, direction and project title.* | *PRPs question what tapering means in the context of the study. They do not like using words such as tapering or withdrawal when discussing tapering.* Both PRPs suggest ways to refine the question and *agree with the final direction and project title.* |
| Designing the study | PRPs recommend data collection from **both clinicians and patients**, items to be included in the survey. *They recommend items to include in the screening questionnaire and identify questions to ask patients during the interviews.* | PRPs recommend including both UC and Crohn's **patients** in the sample, a ranking exercise after the interviews, an interprovincial lens, conducting interviews over focus groups and **blinding the literature review results from the members collecting data.** *They recommend items to include in the screening questionnaire and identify questions to ask patients during the interviews.* |
| Developing the study material | **One PRP develops the recruitment flyer, provides questions for both the patient and clinician interview guides, recommends language to be included in the consent, and provides content for the online surveys.** *One PRP develops the interview guides* for clinicians and provides feedback on the patient interview guide. **Both provide feedback on all the study materials.** | *PRPs develop guides* for the focus group and interviews. |
| Participating in the literature search | PRPs propose questions for the search. **PRPs review papers and extract data.** One PRP identifies papers useful to finalize the definition of tapering and inform the research design. | **PRPs are blinded to the results of the review.** |
| Training team members on how to recruit and work with patients | **No role.** | **PRPs conduct a mock session of the focus group.** |
| Finding patients to participate in the study | *PRPs propose platforms and strategies for recruitment.* One PRP was willing to use their connections to identify potential candidates and support recruitment of patients. **One PRP recruits clinician participants.** | *One PRP provides names of potential recruitment platforms.* |
| Data collection | *One PRP conducts interviews of all the clinicians.* | *PRPs conduct the focus group and interviews.* **They influence the group to drop the "ranking exercise" after the first focus group.** |
| Analysis and Reviewing results | *One PRP reviews the coded data of one clinician transcript and shares insights with the group.* | PRPs take on an advisory role during data analysis. *They review the analyzed data and agree that it resonates with what they heard during the data collection process.* |

**Figure 2** Comparative journey maps of PRPs in the PLG and RLG illustrating patient-centredness. PLG, partner led group; PRP, patient research partner; RLG, researcher led group. Key similarities between the groups are emphasized in orange, italicized text. Key differences are emphasized in blue, bold text.

## RESULTS
### Study participants
Fourteen participants were recruited in total for the two project groups from a pool of 29 eligible participants. The main reasons for non-participation were workload issues and health concerns. The majority were 35 years old and over (PLG n=5; RLG n=6); women (PLG n=5; RLG n=5); white (PLG n=4; RLG n=6)); had a PhD or a professional degree (PLG n=3; RLG n=5) and had been involved in POR for over a year (PLG n=6; RLG n=4). Nine (PLG n=3; RLG n=6) felt prepared to contribute to this study and seven (PLG n=3; RLG n=4) indicated they had previously worked with or knew at least one member in their group before this project.

PRPs in both groups were trained in conducting research projects using qualitative methods through the PaCER programme[41–43] or through other education opportunities. All the researchers had qualitative research expertise; some with no IBD-specific knowledge. The clinicians were affiliated with the SPOR IMAGINE Network.[28 44]

### Critical outcomes of PRP engagement: similarities and differences of PRP engagement in the two groups
We present the observation results by the 11 CORE[35] operationalised in our study (table 1). No new outcome

was identified during our analysis. Patient-centeredness was central to all the outcomes of research engagement but we tried to keep the measures independent of each other during synthesis. We gathered information about all stakeholders but mainly focused on the contributions made by the PRPs in this paper. Representative quotes from both groups for each outcome are included to further illustrate the findings (table 2).

### Patient centred
PRPs in both groups took on both advisory and operational roles. They influenced the project wearing dual hats of patients and researchers. The PLG lead took on many operational roles and influenced more project-related decisions than the RLG lead.

PRPs' experience in the researcher-led group influenced the group to conduct a literature review first to finalise the research question. They reviewed articles along with researchers in their group, extracted data and helped their group identify papers useful to finalise the definition of tapering and study design. They also helped their team determine an optimal study design, recommended inclusion/exclusion criteria and data to collect such as duration of biological use, etc., provided a rationale for collecting data from clinicians and patients, identified questions to ask patients during the interviews,

**Table 1** Critical outcomes of research engagement and study measures*

| Critical outcomes | Measures |
|---|---|
| Patient centred | How were PRPs engaged in, and influenced, each stage of research and critical research tasks? |
| Meaningful | Are the research method and outcomes reflective of, and outcomes relevant to, the community and all group members? |
| Team collaboration | What is the group members' comfort level during discussions? Do all group members trust and respect each other? Are all group members clear about their roles on the project? Are PRPs and researchers given an opportunity to gain skills and knowledge in ways that work for them? |
| Understandable | Are study materials patient-friendly, understandable and written in a common/plain language? Are all group members comfortable with the written materials? Evaluate the reading level of the research documents. Was the data presented in an accessible, understandable way to all members? Is the overarching goal, study purpose and research question understandable by everyone? |
| Rigorous | Did the group appropriately integrate PRP suggestions without compromising rigour? |
| Integrity and adaptable | Did PRPs propose any changes to the study design, methods, materials, etc., that were made/not made? If not made, explain why. |
| Legitimate | To what degree was the sample or study population diverse and representative/unbiased? |
| Feasible | Are research goals and methods realistic and feasible? |
| Ethical and transparent | Are all methods ethical, culturally safe and patient-friendly? Is data/privacy protection more patient-centred and/or changed? Is honest transparent communication consistent throughout the project? |
| Timely | Is conduct of research and sharing information with all group members timely? |
| Sustainable | Is there a plan for sharing study findings? What role did PRPs play to disseminate study findings? |

*Adapted from Dillon's *Critical Outcomes of Research Engagement* (CORE) and measures.Dillon EC, Tuzzio L, Madrid S, et al. Measuring the impact of patient-engaged research: how a methods workshop identified critical outcomes of research engagement. *J Patient Cent Res Rev* 2017;4:237–46. 10.17294/2330-0698.1458
PRP, patient research partner.

developed the clinician interview guides based on their experience, recruited and managed clinician recruitment and data collection and reviewed the coding of one of the clinician transcripts. Patient recruitment, data collection from patients and all the data analysis were managed by RLG researchers.

The PRPs in the patient-led group helped their group define 'tapering' for the purpose of their project. Unlike the PRPs in the RLG, they were not involved in the literature review process to avoid any bias in data collection. The patient lead conducted an informal search of the literature and proposed ideas for a project design and approach. The PRPs provided additional thoughts and structure to this design, some inclusion/exclusion criteria, variables to include in the screening questionnaire and strategies to collect data. For example, one PRP influenced the group to conduct a formal literature review simultaneous to the first focus group to have a draft list of attributes from the patient and clinician perspective that was used to inform the study materials and further data collection. The PRPs also developed the interview guides, conducted the interviews, and reviewed and confirmed the final themes.

PRPs in both groups were involved in data interpretation, in the knowledge translation discussions and came up with potential ideas along with their group members to share project findings.

### Meaningful

PRPs in both groups were part of the decision-making processes during all the project stages, resulting in project deliverables that were relevant and meaningful to them and to the other stakeholders in the group. For example, the PRP experience in the RLG helped their group members better understand biologics and what aspects of withdrawal may be important to capture from their perspective. A PRP shared the side effects she faced due to biologics and even though she was in her third year of remission, was not allowed to get off biologics. This conversation contributed to the group discussing the differences in interpretation of 'tapering', the frequency, dosage, side effects and how that might influence the patient experience with biologics. Even though not much was discussed specifically about treatment by the PRPs in the PLG, their experience provided an insight into how others with similar lived experiences may want to participate in the study. A PRP shared her difficulty navigating insurance coverage for biologics between provinces, resulting in decisions about the inclusion/exclusion criteria of their project. The final list of attributes was discussed and finalised with the PRPs in both groups.

### Collaboration

There was a strong sense of teamwork at the start of both group projects. However, collaboration seemed easier in the PLG than in the RLG which could be attributed to

**Table 2** Illustrative quotes for critical outcomes of research engagement from observation of the two project groups

| Critical outcomes of research engagement in both groups | Illustrative quotes |
|---|---|
| *Patient centred*<br>PRPs in the PLG and RLG were engaged in all the stages and critical tasks of their group's qualitative projects, on advisory and operational roles and influenced the stages and tasks they were involved in.<br>The lived experience of the PRPs in the RLG informed more aspects of the project than the PLG PRPs. The PLG lead influenced more project-related decisions than the lead in the RLG. | 'I just wanted to echo that wearing that kind of dual role in this myself, I think there are definitely a lot of overlaps that I can already see from a research perspective as well as a person experiencing it, so I think it'd be interesting to actually have the literature review and focus groups running simultaneously and some of us that are doing the focus group with the patients and then some of us that are doing the literature review at the end, we kind of merge the two together that way you're not biased by what we're finding in either source?' PRP influencing the design<br><br>'…it'd be interesting to hear about the different treatments they've experienced right? Did they first try steroids, or did they first try a special diet, or something else and how did they get to biologics and then things like what factors would influence your decision for example, like recovery time, hospitals stay, and what would make you feel more confident, and what information would you need from your doctor…' PRP influencing the content of the interview guide |
| *Meaningful research*<br>The research process and content were reflective of the shared experience, beliefs and values of the PRPs in both groups and the two project outputs relevant to all study stakeholders in the groups. The patient engagement in both groups resulted in a meaningful research experience for all the stakeholders in the two groups, with the PLG stakeholders more satisfied with their experience. Members of both groups were satisfied with the research outputs. | 'Do we want to try to get from as many provinces as possible, I think it would be good to have that lens. In my experience, I was diagnosed when I was in BC and I'm a resident of Ontario so it was a very complicated process because I was out of province my health insurance was actually like it was done so I couldn't get the coverage for any medications. I think there will be others who may have similar experiences, or maybe different experiences so it'll be interesting to see how that ranges province to province.' PRP influencing the sampling criterion<br><br>'I think there is value add in having both Crohn and UC perspectives at the table when we're doing a focus group. Just from personal experience, my experience was 180 degrees different from what my sister experience so having that kind of dual lens might be helpful.' PRP influencing the sampling criterion |
| *Team collaboration*<br>Both groups were collaborative through the entirety of the qualitative project process. The group leads made significant efforts to ensure that all members had opportunities to contribute, based on their individual strengths and interests. The PLG met more frequently than the RLG, with decisions taken predominately during these meetings, while many decisions were taken by email and during the group meetings in the RLG. Both groups shared their views and insights. There was support for each other and appreciation/acknowledgement of work. Collaboration seemed easier in the PLG than in the RLG. There was a shared understanding of roles and unanimity in the PLG's final decisions, but not always in the RLG. PRPs in both groups had opportunities to gain new skills and learn from their engagement. | 'I'm trying to figure out what people are interested in. What type of role anyone is interested in. I mean each person that's on the team will have a different appetite for how much they want to be updated, and how much do they want to be involved, more communication.' PLG lead exploring stakeholder interest/roles for the project plan<br><br>'I just wanted to chime into and second what X (other PRP) pointed to from a patient perspective, I think this title works in terms of just capturing what we want in, and that is the people that are on biologics…' PRPs collaborating with the group on project title |

**Table 2** Continued

| Critical outcomes of research engagement in both groups | Illustrative quotes |
|---|---|
| *Understandable*<br><br>The PRP and clinician experience served to understand and contextualise 'tapering' of biologics in both groups. PRPs developed or reviewed the study materials using language that was clear, engaging and agreeable to all the group members. The PLG presented their output in lay language while the RLG used more clinical/research language. | 'I think if you use the word changing you're going to get all the people who are being forced to change to biosimilars right now. Like if we put that in our title, I think we're going to get the wrong people.' PRP reviewing the project materials to ensure an understandable project title<br><br>'The clinician … used a lot of terms, especially… medical abbreviations … so when it comes to transcribing those terms, I will be willing to provide the input. I mostly knew what he was saying there, there were a couple that I'm not familiar with, but can bounce that off you (clinician in the group).' PRP reviewing transcripts to ensure understandable data |
| *Rigorous*<br><br>Throughout all stages of the two group projects, group members took collective decisions, balancing scientific evidence with group member insights. The RLG took a more evidence-based approach while designing the project. The two study groups integrated group member suggestions into the design, approach and conduct of the two qualitative projects without compromising project rigour. | 'I think we are reaching the saturation point, plus this individual is similar in demographic that we already have. I think one or two things this individual might say, but is it going to change the whole direction of where our data lies, I doubt it!' Researcher confirming data saturation<br><br>'Tapering (definition) could be: Decrease of dose; Increase of interval between two infusions/injections; Discontinuation; Replacement by a 'lower' medication. I think it will be very important to clearly define these for participants—the attributes important to patients may vary depending on the type of tapering being considered.' PRP influencing project design |
| *Integrity and adaptable*<br><br>The PLG and RLG group members were flexible and continuously improved the project process if the changes were logical, verifiable, rigorous and ethical. Both groups embraced challenges and found new ways of meeting the project objective. PRPs in both groups were involved in interpreting the data and in identifying the final candidate attributes, ensuring that the research findings were appropriate and justifiable. | 'I have not heard of biologic tapering happening, and when I've talked to my GI about moving off the biologic somehow, he's super uncomfortable because from what I understand, and maybe the research has changed, the risk of recurrence is really when people have gone off. So, I think it's really important to understand what is meant by tapering in this context and the research that's available to support tapering.' PRP influencing the group to study tapering in more depth before designing the project<br><br>'I think we need to drop the ranking exercise (based on what was heard during the first focus group), the ranking would be heavily influenced based on the life experiences that the person had, so depending on who's doing the ranking, the ranking could be skewed and I think it would be difficult for it to be representative of a larger population…' PRP suggests dropping the ranking exercise after conducting the first focus group |

Continued

**Table 2** Continued

| Critical outcomes of research engagement in both groups | Illustrative quotes |
|---|---|
| *Legitimate*<br><br>Diverse and experienced PRPs in the two groups brought value into their project's decision-making process and enhanced the understanding of tapering of biologics from the patient perspective. There was diverse representation of project participants in the qualitative projects of both groups, though the sample size was small in the RLG. Both groups considered how bias might impact their recruitment. | 'a lot of the responses (about who can help make the decision about tapering) came back that it would be great information to get from my gastroenterologist. So, it wasn't like … I'd like to go online and do a Google search and get this information right at my fingertips … they wanted someone to relay that information to them.' PRPs informing the group about the needs of diverse project participants<br><br>'just reflecting on the interviews, the categories seem logical to me, I feel it is pretty accurate. I actually like how it comes out, burden of disease, treatment, financial costs, coverage, I like that decision making- they talked about whether they used their healthcare provider or family or who else they might, like other patients' PRP confirming final list of attributes |
| *Feasible*<br><br>Members in both groups took on roles that were feasible for them. Collaboratively, they planned a project design and approach that was feasible to complete within the timeframe, without compromising the quality of the project. Time constraints experienced by the RLG negatively affected recruitment and data collection. | 'For the study itself, due to time constraints and reflecting on the research question, I think we should focus solely on patient perspectives. We will definitely have to kind of brainstorm and look at the research that's been done before, to see what the best kinds of ways, or how it might be best to … gather their perspectives.' PRP discussing the project design<br><br>'I struggle more to find participants for focus groups than for interviews. I think, for the longer part of the projects relying on multiple focus groups, in the world that we're in right now, might be just difficult to accomplish.' PRP influencing the study approach |
| *Ethical and transparent*<br><br>PRPs in both groups collaboratively helped solve ethical dilemmas, and continuously checked assumptions of other group members during recruitment and data collection to ensure data collection materials and tools were transparent. Risks and potential harms to the patient were considered. | 'I think it would be great to have the clinicians conducting the interviews, my question is would the interviewees be made aware of that?' PRP discussing risks and potential harms<br><br>'Are we trying to encourage people to do things that actually go against … clinical care guidelines.' Clinician questions ethics |
| *Timely*<br><br>The PLG was able to complete their project within the stipulated timeframe, while the RLG spent substantial time defining the question, which prevented the group from completing data collection as planned. Members in both groups took collaborative decisions and made relevant changes in a timely manner. | '… do we have the time to also capture (patient blogs), because we're going to be starting the focus groups, we need to analyze we've got to write this thing up and it's all going to be done by the end of September, there's a lot of work there ahead of us, so … I don't think it's wrong to not include personal blogs if everyone agrees…' Researcher discussing feasibility<br><br>'…the point of the project is for the group to design something that reflects their ideas and what is important to them, so I actually think it is more important to get the design right than to get it done (on time).' RLG lead encourages group to spend more time on research question and design |

Continued

**Table 2** Continued

| Critical outcomes of research engagement in both groups | Illustrative quotes |
|---|---|
| *Sustainable*<br><br>The research addressed most members' needs and expectations, resulting in continued participation on the project. One PRP dropped out of the RLG due in part to unmet expectations. The key outputs met all group member's requirements in the two groups. The PLG offered to present project findings at conferences and workshops and considered publishing their engagement experience. Both groups also proposed future research topics. | 'I'm glad I had an opportunity to review some of the literature in detail. I particularly appreciated reading more about dose reduction, dose cycling, and personalized approaches to tapering – I had always considered tapering as 'discontinuing' altogether, so these expanded concepts related to tapering were really neat to consider…' PRP<br><br>'feels good knowing all the members, and how accommodating everyone is to help out with the project.' PLG Lead |

PLG, partner led group; PRP, patient research partner; RLG, researcher led group.

the clear roles that members had in the PLG; a clear plan for team communication; the lead taking on a number of time-intensive tasks and frequent virtual meetings in the initial phases of the project with a full complement of project members.

Email discussions and decisions were more common in the RLG with the RLG meeting 14 times, over the 7 month project period. The PLG met weekly in the first 2 months of the project, for a total of 24 virtual meetings over the project period. Decisions were taken mainly during the meetings attended by most group members, including one clinician. Irrespective of the approach, members in both groups shared ideas and opinions freely and everyone's opinion was valued. There was small talk before and after meetings, and appreciative notes circulated and mentioned during meetings which made all stakeholders—especially the PRPs—feel appreciated.

Many collaborative decisions were taken by both groups during all project phases, impacting the process and results of the projects. For example, leveraging personal experience, a PRP in the PLG pointed out the importance of including both Crohn's disease and ulcerative colitis in the project. There were conflicting opinions about the study design in the RLG due to the lack of clarity around the project question. This caused frustration and disengagement especially among the PRPs. Many respectful dialogues were held to reach a consensus on the study design. The resultant design included data collection from both patients and clinicians. Another example of collaborative work was seen during the development of the study guides in both groups with input from PRPs and clinicians to ensure comprehensive data collection.

Capacity building of PRPs also facilitated collaboration and in turn impacted the results of the projects. For example, training of a PRP in interview facilitation and conducting mock sessions enabled the PRPs to conduct some of the interviews in the RLG and all PLG interviews.

### Understandable
The PRP and clinician experience helped contextualise 'tapering' and how to frame it in a research question. A patient-friendly project title was subsequently discussed and finalised in both groups.

The PRPs in both groups developed or reviewed the study materials using clear, engaging language suitable for their project participants. They also developed or provided feedback on the data collection tools, ensuring information important to ask study participants was included in the guides and that the guides were easy to administer during data collection. For example, a PRP in the PLG simplified a question in the guide from 'What factors would make you feel confident that this is the right decision?' to 'What would you like to know from your healthcare provider to make a decision?' A PRP's query whether people can hypothetically think about going off biologics without actually wanting to go off biologics got the group rephrasing the introduction section of the guide.

The results of the PLG literature review were presented in simple, understandable terms. The PLG lead influenced the presentation of the final list of candidate attributes in lay language. The RLG used more clinical/research language for some of their attributes.

### Rigorous
Group members made collective decisions in all project stages, balancing scientific evidence with PRP insights. The PLG lead presented initial thoughts and other group members built on her ideas. Wearing both a researcher and patient hat, a PRP proposed running focus groups and a literature review simultaneously, blinding one to another to avoid biasing each other. Many members felt this approach would add value and rigour to the project. Subsequently, the patient-led group conducted an unstructured focus group prior to the formal interviews, alongside a literature review, to better grasp the patient perspective of biologics without being influenced by the literature.

The researcher-led group followed a more evidence-based approach, conducting a formal rapid review of the literature and using the results to establish a clear context and foundation for the research question and

project design. A PRP and a clinician provided a clear rationale for capturing data from both clinicians and patients, which was accepted by group members despite the patient-focused scope of the project.

Both groups used best practices when conducting, analysing and reporting the results. The PLG collected data from one focus group and eight interviews and stopped data collection as no new themes were being identified. The RLG collected data from three clinician interviews and two patient interviews. The sample size of patient interviews was not realised as planned.

The analysed data were shared with RLG members iteratively as it was being coded. The transcripts were double coded in both groups. PRPs and clinicians in the two groups validated the themes and ensured nothing was missed or mis-represented.

### Integrity and adaptable

The PLG and RLG group members were flexible and continuously improved the project process if the changes were logical, verifiable, rigorous and ethical. Both groups embraced challenges and found new ways of meeting the project objective. Common challenges included the lack of clarity of the project purpose, unclear definition of 'tapering', obtaining timely ethics approval and identifying project participants. Additionally, the RLG dealt with a PRP withdrawal from the study due to a variety of reasons. All members in this group responded well to the changing situation, spending additional time working through sticking points in the research question and project design.

The PRP insights in both groups strengthened the quality and trustworthiness of the data in real time. For example, a PLG PRP influenced the decision of dropping the 'ranking exercise' after the interviews because they felt that this exercise would be meaningless given the small sample size and varied life experiences of patients.

The RLG PRPs influenced the decision to not incorporate clinician perspectives in the patient interviews as this approach did not capture all the nuances of the patient experience.

Members in both groups were involved in data interpretation and in identifying the final candidate attributes, ensuring that the research findings were appropriate and justifiable.

### Legitimate

Diverse and experienced PRPs in both groups brought value into their projects' decision-making process and enhanced the understanding of the research question from the patient perspective. They also ensured that the approach and data gathered were relevant to them. The PRP and clinician experience in both groups helped decide the screening questions to capture diverse and/or representative patient perspectives. The PLG interviewed patients from many Canadian provinces, with mild, moderate or severe symptoms of either Crohn's disease or ulcerative colitis, with nearly half of them using multiple

biologics. Even though the sample size was small, the RLG patient participants were of different genders and ages, but from the same province, with both using multiple biologics. Thus, a varied group of qualitative project participants from the IBD community provided their perspectives, resulting in an increased understanding about patient preferences for biological medications; the appeal and feasibility of tapering biologics and the perceived benefits and risks of doing so. The qualitative interview findings also confirmed the values of some PRPs about the importance of shared decision-making with their gastroenterologist and other healthcare providers on tapering biologics

### Feasible

Managing the workload on top of full-time jobs or coursework, and other responsibilities, especially within short timeframes, was challenging for many members in both groups. The leads took on many time-consuming tasks, making it easier for group members to participate in the research. Both groups discussed and debated the feasibility of various project designs and collaboratively came up with solutions to accomplish the project goal. Influenced by a PRP and clinician, the RLG decided to conduct clinician and patient interviews to address the complex study question. The PLG also discussed various designs, but decided to interview patients to complete the project within the timeframe. The PLG lead, and the RLG PRPs leveraged past connections within the IBD community to help with recruitment.

### Ethical and transparent

Each group collaboratively solved ethical dilemmas such as privacy of participants during the zoom focus group sessions versus individual interviews; how to start an interview so that participants feel safe and secure to discuss their personal experiences; whether to put the honorarium amount on the recruitment flyer; how to recruit participants without putting them and the PRPs at risk, etc. PRPs as well as researchers in both groups were sensitive to ethics practices, particularly surrounding recruitment and data collection.

For example, the PLG did not recruit through gastroenterology clinics since gastroenterologists were not particularly interested in discussing biological tapering with their patients. All information was transparent to group members in the recruitment materials shared with them. The whole process (the methodology, including the design, data collection, coding, analysis and tools used in data analysis) was discussed and known to all members of both groups. PRPs in the RLG specifically shared the qualitative study results with their group participants to satisfy the goal of transparency. Engagement during analysis ensured that the patient perspective was transparent in the findings.

### Timely

To ensure timeliness of the two projects, both leads took on many tasks. Virtual meetings and finding convenient

times were a hurdle; the PLG ended up scheduling late-evening meetings. Group commitment to project success, sharing of responsibilities and an interest in POR kept the two projects moving forward.

A feasible design enabled recruitment and data collection in the PLG. Prompt responses and constructive PLG meetings also contributed to timely decision making. Revisiting the project plan periodically was also helpful. The RLG spent substantial time during the early stages of the project building collaboration and coming to a consensus about the project question, which hindered the project timeline. The RLG lead recognised that completion of the project on time was important, but secondary to ensuring that all group members were happy with the research question and project design.

## Sustainable

PE was visible throughout the research process and across various research tasks in both groups. All group members, including PRPs, had the necessary prerequisites (training, exposure and preparedness) for making decisions and engaging on the project until the end. No health episodes prevented sustained engagement. Immediately defining and distributing group member roles significantly contributed to the sustainability of the PLG project. The design, approach, materials and results eventually met the needs and expectations of all members in both groups, resulting in continued participation. Beyond sharing the results through publications and presentations, the two groups proposed future research topics such as developing and evaluating decision aids for shared decision making.

Our results show that engagement of PRPs in research has a positive influence on the research design and delivery in the context of qualitative research in both the patient-led and researcher-led group. Using their lived experience, research knowledge and other life skills and experiences, PRPs in both groups helped operationalise the research question, the project design and approach; conducted or participated in the literature review; collected data and analysed data or provided input in the analysis and interpretation of the results. During the initial stages of the project, the PRPs in the RLG influenced their group to conduct a literature review first before finalising the design. The PRPs in the PLG influenced their group to conduct the formal review simultaneously with the first focus group. They used the information to develop their study materials as well as during data analysis. The PRPs in the RLG influenced their group to collect information from both clinicians and patients. The PLG collected information from only patients. The final list of attributes was reviewed and finalised with the PRPs in both groups. As such, the research and the list of attributes were relevant and reflective of the lived experience, beliefs and values of the PRPs in both groups. The stakeholders valued the experiences and knowledge that PRPs brought to the group. The resultant projects were patient-centred, collaborative, meaningful, rigorous, adaptable, ethical, legitimate, understandable, feasible, timely and sustainable.

## DISCUSSION

Using observation, we comprehensively measured the outcomes of engagement across the research spectrum and obtained contextualised insights of engagement in the two groups. We gained a better understanding of the key ingredients to successful engagement; the influence PRPs had on the research and operationalised the CORE. For example, we observed how the working partnership ensured transparency or fairness in the projects or what changes PRPs proposed that were made/not made and why. We also identified ways the two groups appropriately integrated group member suggestions without compromising project rigour. This study enriches existing literature using the observation method to assess research engagement, teasing out the input and influence of PRPs. While previous research has used methods such as surveys, interviews and focus groups to study engagement, the current study demonstrates that observation can be an effective method, provided the expertise to conduct and record the observations and resources are available.[45]

The PRP experience on the projects was not tokenistic[46]—they engaged in multiple ways across the research phases ranging from sharing their experience to coproducing research.[47–50] No power imbalances[51 52] were observed. Members shared ideas with each other throughout the project. Informed decisions were made jointly through discussions. Small talk at the start and close of meetings, positive and encouraging feedback from the researchers and clinicians also made PRPs feel appreciated. These qualities are essential to nurture interpersonal relationships between group members.[52] Consistent with emerging literature, our results demonstrate that engagement can be sustained across the research spectrum and not limited to preliminary activities,[53] provided there is adequate preparation and resources (ie, funding, time);[54] motivation at both the patient and researcher level;[3] training and supports for researchers to effectively engage with patients;[3 54] adequate training and supports of PRPs[54] and willingness of PRPs to take on roles in the later stages of a project.[55] PRPs in both groups had high-level skills and training in POR and/or qualitative research, and could function both as researchers and patients, which is unusual in health research. Some researchers also wore dual researcher and patient or researcher and clinician hats. The PLG acknowledged their dual roles and identities through 'reflexivity'.[56] Studies have shown simple acknowledgement is insufficient, but concrete reflexive practices can help build trust, ensure transparency, authenticity and more rigorous research.[57 58]

The group leads were also vital in promoting engagement.[5] Previous studies suggest that the leads could be the main stumbling blocks to engagement if they lack the knowledge, skills and experience on how best to do it and do not possess the leadership qualities for collaborative work.[59 60] Our group leads were organised, communicative, respectful and committed and regularly checked in or provided updates to group members. They 'led' the operations of the project and 'facilitated' engagement.

We also observed that relationship-building with PRPs in research takes time[61] and includes: a flexible engagement plan with clarity about roles and expectations, clarity about the purpose and format of the collaboration, agreed goals, agreed communication strategies and ways to monitor project progress.[3 47 62–65] Core values that the diverse members bring to projects should also be discussed for successful engagement, such as mutual respect and trust, equal partnerships, appreciation, compromise and support for each other.[66–68]

Our study was exploratory and would be difficult to replicate since it is not possible to control the myriad characteristics of the group members and the context. Furthermore, operationalising these outcomes was challenging as they were established for direct inquiry with study team members, with overlapping measures among the 11 outcomes. However, the findings of the study offer important insights into the value of engaging with PRPs in the context of patient preference studies. Future research using the observation methodology to examine outcomes of research engagement in other contexts and settings requires appropriate resourcing, and careful design to adequately address associated methodological challenges of observing and reporting engagement.

**Author affiliations**
[1]Department of Community Health Sciences, University of Calgary, Calgary, Alberta, Canada
[2]IMAGINE SPOR Chronic Disease Network, Hamilton, Ontario, Canada
[3]The University of British Columbia School of Population and Public Health, Vancouver, British Columbia, Canada
[4]Michael Smith Health Research, Vancouver, British Columbia, Canada
[5]British Columbia SPOR SUPPORT Unit, Vancouver, British Columbia, Canada
[6]Alberta SPOR SUPPORT Unit, Calgary, Alberta, Canada
[7]Department of Paediatrics, University of Calgary and Alberta Health Services, Calgary, Alberta, Canada
[8]McMaster University Faculty of Health Sciences, Hamilton, Ontario, Canada

**Acknowledgements** The authors thank the 14 group members who participated in this study. They also gratefully acknowledge the support from study team members Aida Fernandes, Executive Director IMAGINE Network, Tracy Wasylak, Chief Program Officer, Strategic Clinical Networks with Alberta Health Services, Dr Gilaad Kaplan, Gastroenterologist and Professor in the Cumming School of Medicine at the University of Calgary and Louise Morrin, Senior Provincial Director, Medicine Strategic Clinical Network at Alberta Health Services during the different study phases. The authors also express their appreciation for the funders of this study.

**Contributors** DAM, the guarantor, conceptualised the study and led the design, conduct and analysis of this study and the drafting of and revising of the article. DAM, DCL and SB conceptualised the study and led the design, conduct and analysis of this study and helped revise the manuscript. NS and KLB participated in the design, coordination, data collection, conduct and analysis of the study and in drafting and revising the manuscript. PM contributed to the acquisition and interpretation of data and reviewed the manuscript critically. GM, SZ, TLM and MJS participated in the design, conduct and analysis of the study and reviewed the manuscript critically. All authors approved the final version to be published and agreed to be accountable for all aspects of the work.

**Funding** This work was supported by the SPOR IMAGINE (Strategy for Patient-Oriented Research, Inflammation, Microbiome and Alimentation: Gastro-Intestinal and Neuropsychiatric Effects) Network. The Network is supported by a grant from the Canadian Institute of Health Research (Funding Reference Number: 1715-000-001) with funding matched by McMaster University, University of Calgary, University of Alberta, Queen's University, Dalhousie University, Montreal Heart Institute Research Centre, Takeda Pharmaceutical Company, Allergan Incorporated, Alberta Innovates, Research Manitoba, and Crohn's and Colitis Canada.

**Competing interests** DAM discloses consulting fees from the Office for Health Economics, Novartis and Analytica during the conduct of this study. She also received support from Illumina for travel expenses to attend a meeting. NS and KLB received reimbursement of expenses related to conference attendance from the SPOR IMAGINE Chronic Disease Network. All other authors declare no conflicts of interest relevant to the content of this article.

**Patient and public involvement** Patients and/or the public were involved in the design, or conduct, or reporting or dissemination plans of this research. Refer to the Methods section for further details.

**Patient consent for publication** Not required.

**Ethics approval** This study involves human participants. All relevant ethics approvals were obtained prior to data collection from the University of Calgary (REB20-1563) and the University of British Columbia (H20-03385). Participants gave informed consent to participate in the study before taking part.

**Provenance and peer review** Not commissioned; externally peer reviewed.

**Data availability statement** No data are available. The ethics approval for this study does not support the sharing of raw data.

**ORCID iDs**
Deborah A Marshall http://orcid.org/0000-0002-8467-8008
Nitya Suryaprakash http://orcid.org/0000-0001-8032-9129
Danielle C Lavallee http://orcid.org/0000-0002-5555-9675
Karis L Barker http://orcid.org/0000-0002-3530-566X
Sandra Zelinsky http://orcid.org/0000-0001-5531-7660
Tamara L McCarron http://orcid.org/0000-0001-7242-1910
Maria J Santana http://orcid.org/0000-0002-0202-5952
Paul Moayyedi http://orcid.org/0000-0002-3616-9292
Stirling Bryan http://orcid.org/0000-0001-7093-3058

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
