## [Reviewer comments · BMJ Open]

ARTICLE DETAILS

TITLE (PROVISIONAL)	Exploring the outcomes of research engagement using the observation method in an online setting
AUTHORS	Marshall, Deborah; Suryaprakash, Nitya; Lavalley, Danielle; Barker, Karis; Mackean, Gail; Zelinsky, Sandra; McCarron, Tamara; Santana, Maria-Jose; Moayyedi, Paul; Bryan, Stirling

VERSION 1 – REVIEW

REVIEWER	Wenke, Rachel Gold Coast Hospital and Health Service, Allied Health
REVIEW RETURNED	30-May-2023

GENERAL COMMENTS	Thank you for the opportunity to review this research which explores an important area regarding the dynamics of research collaborations that involve researchers and patients. Involving patients (or consumers) in research is a topical and relevant issue to researchers and your paper provides useful insights into the practicalities of doing so. The study is well written however I have some minor comments below to assist with the overall readability. Abstract: Conclusion- this is a very broad and general statement, can you make more specific how "engagement" leads to positive outcomes and the nature of engagement so this statement is more meaningful to the reader regarding implications of this research. The introduction is well written and sets the scene for the rationale for this study. Methods: The use of the acronyms PLG, RLG, PRP is reduced readability throughout the document. I would consider simplifying and spelling out where possible. p6- when describing "the data from both groups were analysed thematically..." does this also include the observations and reflections from the observer? If not, how were these analysed? Results: When describing the demographics of the groups it may be useful to separate the differences in the amount of PhDs between the two groups as this may reflect the level of research experience between members. Figure 2 is useful at summarising the similarities and differences between the groups.
--

	P17 it is unclear in the sentences following the statement "Collaboration seemed easier in the PLG" are referring to the PLG or both groups? If it is just the PLG - was there any evidence of body cues indicating uneasiness in the RLG? P21- Table 2- can you provide justification why you didn't include in the quotes which group the quote belonged to? Discussion: Can you provide any more specific suggestions for future research in this area? What should additional funding and resources be used towards?
--	---

REVIEWER	Hollin, Ilene L. Temple University
REVIEW RETURNED	10-Jun-2023

GENERAL COMMENTS	This study aims to examine the impact of patient engagement on a qualitative study about IBD. They utilized an observational approach to collect data about the impact of patient engagement. This is interesting work with great potential to help understand the impact of patient engagement and the ways it can be used effectively, however there concerns with the presentation of the work that weakened this potential. These key issues are outlined below. Major/primary concerns: First, there appears to be a disconnect between the objective (to compare the type of patient engagement that occurs within a RLG and a PLG) and the discussion/conclusions which seem to focus on patient engagement more broadly. The reader is left wondering if there is a benefit to a PLG? Or are the results the same or better with a RLG? Second, how did the difference ways in which the two groups conducted their respective qualitative research studies ultimately impact the results of the qualitative study. If patient engagement in the form of RLG gets to the same place at the end of the qualitative research study as PLG does, then one wonders, is there an added benefit to PLG? I am not clear from this article what the authors perceive to be the benefit, if any, to PLG. Abstract: The abstract should contain information about the context in which this research was conducted (what disease, population, etc). Poorly written, shows lack of attention to detail in terms of sentence construction, inconsistent and incorrect use of capitalization and incomplete sentences. Use of term "projects" to refer to "groups" is confusing. I would consider this one project with two groups. The abstract is written such that it appears as if the results indicate that the RLG had richer data than the PLG, or at least that the PRPs had a greater impact in the RLG than the PLG. However, the discussion in the body of the article does not necessarily reflect that. Note: Page numbers below refer to the page number indicated at the top of the PDF, where it says page X of 44 (not the page number in the bottom right corner, which is different). Intro
--

	Page 5, Lines 12-17: Values listed here sounds like values of engagement for patients, not for researchers, but the sentence mentions researchers in line 12. Page 6, Lines 3-8: The objective is not clear. This sentence fails to mention the comparison between the PLG and RLG, which is a main component of what it appears the authors aim to do (based on the abstract). Similar to the abstract, the intro provides no context in terms of what type of patients/diseases/situations the authors are dealing with here. It is not until page 6 (methods) that I understand this is about IBD patients. Further, there is no context regarding what is the aim of the qualitative study that the two groups conducted? Methods Page 6, Line 35 – The acronym POR is used without defining it. Page 6, Line 47 – Matching two groups based on what? Page 6, Line 50- Page 7, Line 17 – This whole paragraph belongs in the intro as it is not the methods of the current study which is a comparison of two forms of engagement. The authors are conflating the methods of the qualitative research study about attributes with the qualitative research study about the impact of engagement. The methods section of this paper should reflect the methods relevant to the research study about the impact of engagement only. Everything else is relevant context that better belongs in the intro/background. Generally, the methods section lacks sufficient details. Some examples are: Page 7, Line 52-Page 8, Line 3: Who reviewed the audio recordings? A second staff person or the same staff person? If the same person, then the audio recordings could not verify quality or trustworthiness, but perhaps only accuracy. If a different person, you could address concordance. Page 8, Line 4-6: What received a study ID number? An interaction? A meeting only? Page 8, Line 26-29: unclear sentence Page 9, Figure 1: Step 1: Transcription, anonymization and importing transcripts is not organizing data. These are critical steps to prepare the data for analysis. I suggest calling this "Data Preparation". I would include importing the transcripts under step 1. Step 2: This step should be coding and should include coding, modifying the code book and then RECODING. Recoding after modifying code definitions is important and appears to be overlooked. Step 3: What is the difference between the first and second bullet? Page 9, Lines 30-35: How did the coders perform quality checks? What was the discussion? How was quality checked? Were codes compared to see if two study staff coded the data the same? Did 2 study staff each code all the data and compare to see if codes were applied the same? Or did 1 staff member code half the data and 1 staff member code the other half? Page 9, Line 38: How were reflections captured and recorded? How were biases used? Page 9, Line 49: What is meant by high-level information about project group work and deliverables? It would seem that a project lead of a qualitative research study on attributes would need to know all the details about the work and deliverables. Results and Discussion
--	--

	Page 10, Line 49 – This comment belongs in the discussion, not results. Page 11, Table 1 (and the introduction of core at the bottom of page 10) belongs in the methods as this is a tool used for analysis, but not results itself. Overall, the results don't provide enough evidence or examples of the work. For example, on page 12, Line 37, show the reader how the the deeper understanding of needs informed the design, approach and conduct. What specific decisions were made that reflected this? Page 13, Line 3-8: Were the materials that were developed differently across the groups considerably different in the end? Were the materials from one group better than the other? The results lack usable, relevant information. Much of the results points to similarities or differences that lack specificity, tangible examples and understandable information about how those differences ultimately impacted the qualitative study of attributes. Within each results section, there appears to be a long list of similarities and differences, but it lacks cohesion about what the long list taken together means. In other words, the manuscript is missing the answer to "so what?"
--	---

VERSION 1 – AUTHOR RESPONSE

Reviewer: 1

Dr. Rachel Wenke, Gold Coast Hospital and Health Service

Comments to the Author:

1) Thank you for the opportunity to review this research which explores an important area regarding the dynamics of research collaborations that involve researchers and patients. Involving patients (or consumers) in research is a topical and relevant issue to researchers and your paper provides useful insights into the practicalities of doing so. The study is well written however I have some minor comments below to assist with the overall readability.

Author response: Thank you for your positive review and constructive feedback. We are keen to move the science of patient engagement forward and share our learnings and insights with others conducting research in this field. We appreciate your suggestions that improve overall readability.

2) Abstract: Conclusion- this is a very broad and general statement; can you make more specific how "engagement" leads to positive outcomes and the nature of engagement so this statement is more meaningful to the reader regarding implications of this research.

Author response: We have provided more context to our conclusion statement to make it more meaningful to the reader.

“Our findings show that engagement of PRPs in research has a positive influence on the research design and delivery in the context of qualitative research in both patient-led and researcher-led groups.”

3) The introduction is well written and sets the scene for the rationale for this study.

Author response: Thank you, we are glad that you found the introduction sufficiently set the scene for the rationale for our study.

Methods:

4) The use of the acronyms PLG, RLG, PRP is reduced readability throughout the document. I would consider simplifying and spelling out where possible.

Author response: We used acronyms throughout the document to reduce the overall word count. We are happy to expand these acronyms throughout the manuscript or at the start of each section, or include them occasionally in the results section as we have currently done to improve the readability of the document. We defer to the editor for advice on this since it will have implications for word count.

5) p6- when describing "the data from both groups were analyzed thematically..." does this also include the observations and reflections from the observer? If not, how were these analyzed?

Author response: Yes, the descriptive and the reflective data from the observer were both analyzed thematically along with the transcript of the observation. We include text to reflect that:

"The data (transcripts, descriptive and reflective notes) from both groups were analyzed thematically..."

Results:

6) When describing the demographics of the groups it may be useful to separate the differences in the amount of PhDs between the two groups as this may reflect the level of research experience between members.

Author response: We now only include the number of PhDs and professional degrees in each group:

"The majority were 35 years old and over (PLG n=5; RLG n=6); women (PLG n=5; RLG n=5); white (PLG n=4; RLG n=6)); had a graduate, PhD or a professional degree (PLG n=5; RLG n=7); **had a PhD or a professional degree (PLG n=3; RLG n=5)**"

7) Figure 2 is useful at summarising the similarities and differences between the groups.

Author response: We are glad you found Figure 2 useful for summarizing the similarities and differences between the two groups.

8) P17 it is unclear in the sentences following the statement "Collaboration seemed easier in the PLG" are referring to the PLG or both groups? If it is just the PLG - was there any evidence of body cues indicating uneasiness in the RLG?

Author response: We are referring to only the Patient-led- group (PLG). We provide more details that helped shape and support the statement.

"However, collaboration seemed easier in the PLG than in the RLG which could be attributed to the clear roles that members had in the PLG; a clear plan for team communication; the lead taking on a number of time-intensive tasks; and frequent virtual meetings in the initial phases of the project with a full complement of project members."

9) P21- Table 2- can you provide justification why you didn't include in the quotes which group the quote belonged to?

Author response: We did not mention the group associated with each quote to protect participant identities, given that we have only 2 PRPs in each group.

Discussion:

10) Can you provide any more specific suggestions for future research in this area? What should additional funding and resources be used towards?

Author response: We are proposing future research examining the outcomes of research engagement in other contexts and settings using the observation methodology. This methodology requires appropriate resources and funding to conduct and report. Please see our revised sentence below.

“Future research using the observation methodology to examine outcomes of research engagement in other contexts and settings, requires appropriate resourcing, and careful design to adequately address associated methodological challenges of observing and reporting engagement.”

Reviewer: 2

Dr. Ilene L. Hollin, Temple University

Comments to the Author:

This study aims to examine the impact of patient engagement on a qualitative study about IBD. They utilized an observational approach to collect data about the impact of patient engagement. This is interesting work with great potential to help understand the impact of patient engagement and the ways it can be used effectively, however there are concerns with the presentation of the work that weakened this potential. These key issues are outlined below.

1) Major/primary concerns: First, there appears to be a disconnect between the objective (to compare the type of patient engagement that occurs within a RLG and a PLG) and the discussion/conclusions which seem to focus on patient engagement more broadly. The reader is left wondering if there is a benefit to a PLG? Or are the results the same or better with a RLG? Second, how did the difference ways in which the two groups conducted their respective qualitative research studies ultimately impact the results of the qualitative study. If patient engagement in the form of RLG gets to the same place at the end of the qualitative research study as PLG does, then one wonders, is there an added benefit to PLG? I am not clear from this article what the authors perceive to be the benefit, if any, to PLG.

Author response: We are pleased that you see great potential in our study and hope that the revisions we made in response to your constructive feedback clarify your concerns.

In response to your first concern, our study was not designed to arrive at conclusions that one group is better than the other. Our intention was not to judge the leads or the groups, but to look more broadly at how patient-research-partners engage in and influence the research, and report the outcomes of engagement from two groups using each of the 11 domains of the Critical Outcomes of Research Engagement (CORE) framework and identify any new domains that might be relevant. Our rationale for studying two groups (representing two different “levels” of engagement) was to assess PE in two similar but distinctly different groups where PRPs would have sufficient opportunities to contribute and participate in the governance and decision making across all stages of the research.

In response to your second concern, the two groups used different approaches to conduct their qualitative research studies resulting in a list of attributes that shared some similarities but were different. Members in both project groups, including the PRPs, were satisfied with

the final list of attributes identified by their group. Our findings indicate that irrespective of who leads the project groups, PRP engagement in research is beneficial.

Abstract:

2) The abstract should contain information about the context in which this research was conducted (what disease, population, etc).

Author response: We have provided this context in the revised abstract under the study design.

Design: "We observed engagement in two groups comprised of patients, clinicians and researchers tasked with conducting a qualitative preference exploration project in IBD..."

3) Poorly written, shows lack of attention to detail in terms of sentence construction, inconsistent and incorrect use of capitalization and incomplete sentences.

Author response: We addressed these editorial errors in our abstract.

Participants: Patient research partners (n=5), researchers (n=5), and clinicians (n=4) participated in the study.

Main outcome measures: Transcripts of meetings, descriptive and reflective observation data of engagement during meetings and email correspondence between group members were analyzed to identify the outcomes of PE.

Results: Both projects were patient-centered, collaborative, meaningful, rigorous, adaptable, ethical, legitimate, understandable, feasible, timely and sustainable. Patient research partners (PRPs) in both groups wore dual hats as patients and researchers and influenced project decisions wearing both hats. They took on both advisory and operational roles. Collaboration seemed easier in the PLG than in the RLG. The RLG PRPs spent more time than their counterparts in the PLG sharing their experience with biologics and helping their group identify a meaningful project question. A formal literature review informed the design, study materials and analysis in the RLG while the formal review informed the study materials and analysis in the PLG. A PRP in the RLG and the PLG lead leveraged personal connections to facilitate recruitment. The outcomes of both groups were meaningful to all members of the group.

Conclusions: Our findings show that PE has a positive influence on the research design and delivery in the context of qualitative research in both the patient-led and researcher-led groups.

4) Use of term "projects" to refer to "groups" is confusing. I would consider this one project with two groups.

Author response: We agree that the use of these two terms can be confusing. This was a deliberate decision taken by the study team during the initial phase of the research to help differentiate the main project from the two sub projects. We have added a sentence right at the beginning of the methods section indicating that we have called our overall research "the study", and the group work as "projects" and have been consistent in using this terminology throughout the paper. We hope this will help address any confusion.

5) The abstract is written such that it appears as if the results indicate that the RLG had richer data than the PLG, or at least that the PRPs had a greater impact in the RLG than the PLG. However, the discussion in the body of the article does not necessarily reflect that.

Author response: We have revised our abstract so that it does not send the message that the RLG had richer data or had a greater impact. The RLG PRPs spent more time than the PLG PRPs discussing their experience with biologics. These discussions influenced the projects differently. For example, In the RLG, these discussions resulted in defining “ tapering” while in the PLG, these discussions resulted in defining the inclusion/exclusion criteria of their project.

“The RLG PRPs spent more time than their counterparts in the PLG sharing their experience with biologics and helping their group identify a meaningful project question. A formal literature review informed the design, study materials and analysis in the RLG while the formal review informed the study materials and analysis in the PLG. A PRP in the RLG and the PLG lead leveraged personal connections to facilitate recruitment. The outcomes of both groups were meaningful to all members of the group.”

6) Note: Page numbers below refer to the page number indicated at the top of the PDF, where it says page X of 44 (not the page number in the bottom right corner, which is different).

Author response: We have revised to ensure that page numbers are included once at the bottom, right corner of each page.

Intro

7) Page 5, Lines 12-17: Values listed here sounds like values of engagement for patients, not for researchers, but the sentence mentions researchers in line 12.

Author response: The values listed are for patients and not researchers. We have deleted researchers from this line.

“There are also studies showing the value of such engagement to the patient”

8) Page 6, Lines 3-8: The objective is not clear. This sentence fails to mention the comparison between the PLG and RLG, which is a main component of what it appears the authors aim to do (based on the abstract).

Author response: As mentioned earlier, our intention was not to judge the leads or the groups, but to look more broadly at how patient-research-partners engage in and influence the research, and report the outcomes of engagement from two groups using each of the 11 domains of the Critical Outcomes of Research Engagement (CORE) framework and identify any new domains that might be relevant. Our rationale for studying two groups (representing two different “levels” of engagement) was to assess PE in two similar but distinctly different groups where PRPs would have sufficient opportunities to contribute and participate in the governance and decision making across all stages of the research.

9) Similar to the abstract, the intro provides no context in terms of what type of patients/diseases/situations the authors are dealing with here. It is not until page 6 (methods) that I understand this is about IBD patients. Further, there is no context regarding what is the aim of the qualitative study that the two groups conducted?

Author response: We addressed this concern by moving a paragraph from method section that includes the context, aim of the qualitative study that the two groups conducted etc. to the introduction.

“Both groups designed and conducted an exploratory qualitative preference project over a pre-determined seven-month period, addressing the same research question: “What factors or attributes are important to patients with Inflammatory Bowel Disease (IBD) in considering treatment tapering of biologics?” We used this question as the context for studying the impact of engagement since there is no standard regimen for managing adults with IBD and little

evidence on patient preferences regarding treatment decisions when considering biologic tapering. Moreover, the engagement of patients in the development and design of preferences studies is recommended as good research practice.”

Methods

10) Page 6, Line 35 – The acronym POR is used without defining it.

Author response: Thank you for bringing this to our attention. We define it now:

“PRPs and researchers were eligible to participate if they had basic knowledge and skills to conduct qualitative research acquired either through patient-oriented-research (POR) training, education or participating in health care research.”

11) Page 6, Line 47 – Matching two groups based on what?

Author response: We have inserted these details. The two groups were matched by their POR and qualitative research experience and training, and demographics.

“...matching the two groups to the extent possible by their POR and qualitative research experience and training, and demographics.”

12) Page 6, Line 50- Page 7, Line 17 – This whole paragraph belongs in the intro as it is not the methods of the current study which is a comparison of two forms of engagement. The authors are conflating the methods of the qualitative research study about attributes with the qualitative research study about the impact of engagement. The methods section of this paper should reflect the methods relevant to the research study about the impact of engagement only. Everything else is relevant context that better belongs in the intro/background.

Author response: Thank you, we moved this paragraph to the introduction section.

Generally, the methods section lacks sufficient details. Some examples are:

13) Page 7, Line 52-Page 8, Line 3: Who reviewed the audio recordings? A second staff person or the same staff person? If the same person, then the audio recordings could not verify quality or trustworthiness, but perhaps only accuracy. If a different person, you could address concordance.

Author response: We have revised this section and provided additional details including details in the example above.

“All group meetings were audio recorded to verify observation notes and transcripts for their accuracy, quality and trustworthiness. The two staff listened to their group’s recordings to ensure the transcripts were verbatim and their descriptive and reflective notes captured the non-verbal cues, the pre-defined themes and quotes accurately. A third staff performed oversight of this work at various points in the study and resolved discrepancies.”

14) Page 8, Line 4-6: What received a study ID number? An interaction? A meeting only?

Author response: We clarify that the transcripts of meetings, descriptive, reflective notes of the meetings and emails between study team members were anonymized with the participants unique study numbers.

“Ethical practices were followed such as assigning a unique study number on all the transcripts of meetings, emails and descriptive and reflective notes”

15) Page 8, Line 26-29: unclear sentence

Author response: We have revised this paragraph to make our process clearer to the reader.

“The data (transcripts of meetings, descriptive and reflective notes) from both groups were analyzed thematically in four steps using NVivo-12 software: 1) prepared and organized the data for analysis; 2) coded the data by critical outcomes, research stages and critical activities; 3) created a journey map for each group member by “member types” (PRPs, researchers and clinicians) to understand how each member type influenced and impacted the project; **and 4) compared the journey maps of all stakeholders especially the PRPs to identify the overall critical outcomes of PRP engagement in research.**”

16) Page 9, Figure 1:

Step 1: Transcription, anonymization and importing transcripts is not organizing data. These are critical steps to prepare the data for analysis. I suggest calling this "Data Preparation". I would include importing the transcripts under step 1.

Step 2: This step should be coding and should include coding, modifying the code book and then RECODING. Recoding after modifying code definitions is important and appears to be overlooked.

Step 3: What is the difference between the first and second bullet?

Author response: We revised Figure 1 to address all three points.

17) Page 9, Lines 30-35: How did the coders perform quality checks? What was the discussion? How was quality checked? Were codes compared to see if two study staff coded the data the same? Did 2 study staff each code all the data and compare to see if codes were applied the same? Or did 1 staff member code half the data and 1 staff member code the other half?

Author response: Thank you for these questions. We address each in the updated text below:

“Data collection and analysis proceeded simultaneously using the CORE as a priori framework. Two study staff (NS and KB) coded their group data independently. A third coder staff (GM) coded some data from both groups at different stages of the project, merged their coding with NS or KB, discussed discrepancies and reached an agreement on the codes, subcodes and their descriptions. Updated versions of the coding frame were shared between the two staff via the third staff and the data was recoded. After data collection was complete, the two staff created journey maps by stakeholder type for their respective groups. The staff reviewed the journey maps of both groups, and revisited the coding done to ensure

that both agreed on the final journey maps. The journey maps of the patient-led group were compared to the research-led group maps to arrive at the final list of outcomes of research engagement. We also held a virtual meeting with each group separately as a “member check-in exercise” to verify their results.”

18) Page 9, Line 38: How were reflections captured and recorded? How were biases used?

Author response: Both descriptive and reflective notes were captured on a word document and discussed with study team members during meetings to guide further data collection and generation. We now include details about how reflections were captured and recorded, and how biases were addressed.

“Due to the COVID-19 pandemic, and the location of group members, observation of engagement was virtual. We assigned one study staff (NS and KB) per group, skilled in qualitative research, to observe unobtrusively, documenting all exchanges of online meetings and emails among group members. The staff kept notes using a semi-structured guide³¹ of the number of people involved in the discussions, the date of the discussion and the interactions and behaviors between group members (descriptive data). They also recorded their thoughts, questions, biases, initial interpretations of the discussions, potential themes, and direct quotes that seemed significant on a word document (reflective data) (Supplementary Table 1). These notes were discussed during study team meetings to guide further data collection and generation.”

19) Page 9, Line 49: What is meant by high-level information about project group work and deliverables? It would seem that a project lead of a qualitative research study on attributes would need to know all the details about the work and deliverables.

Author response: The group leads were provided high-level information about DCEs, qualitative research and about the deliverables. We decided to remove the word “high-level” to avoid any confusion and have revised this para to make it simpler and clearer to the reader.

“We provided the two group leads training about patient preference studies, qualitative research and about the project group work and deliverables. All this information was made available for use by other members of the two groups.”

Results and Discussion

20) Page 10, Line 49 – This comment belongs in the discussion, not results.

Author response: We have moved this comment to the discussion section.

21) Page 11, Table 1 (and the introduction of core at the bottom of page 10) belongs in the methods as this is a tool used for analysis, but not results itself.

Author response: Thank you, Table 1 is now included in the methods section.

21) Overall, the results don't provide enough evidence or examples of the work. For example, on page 12, Line 37, show the reader how the deeper understanding of needs informed the design, approach and conduct. What specific decisions were made that reflected this?

Author response: We have revised our results section and have provided examples as evidence to show how PRPs influenced decisions.

For example, under outcome meaningful,

“ PRPs in both groups were part of the decision-making processes during all the project stages, resulting in project deliverables that were relevant and meaningful to them and to the other stakeholders in the group. For example, the PRP experience in the RLG helped their group members better understand biologics and what aspects of withdrawal may be important to capture from their perspective. A PRP shared the side effects she faced due to biologics and even though she was in her third year of remission, was not allowed to get off biologics. This conversation contributed to the group discussing the differences in interpretation of “tapering”, the frequency, dosage, side effects and how that might influence the patient experience with biologics. Even though not much was discussed specifically about treatment by the PRPs in the PLG, their experience provided insight into how others with similar lived experiences may want to participate in the study. A PRP shared her difficulty navigating insurance coverage for biologics between provinces, resulting in decisions about the inclusion/exclusion criteria of their project. The final list of attributes in both groups were discussed and finalized with the PRPs in both groups.”

22) Page 13, Line 3-8: Were the materials that were developed differently across the groups considerably different in the end? Were the materials from one group better than the other? The results lack usable, relevant information. Much of the results points to similarities or differences that lack specificity, tangible examples and understandable information about how those differences ultimately impacted the qualitative study of attributes.

Within each results section, there appears to be a long list of similarities and differences, but it lacks cohesion about what the long list taken together means. In other words, the manuscript is missing the answer to "so what?"

Author response: We have included a paragraph at the end of the results section summarizing the results and providing the “so what”

“Our results show that engagement of PRPs in research has a positive influence on the research design and delivery in the context of qualitative research in both the patient-led and researcher-led groups. Using their lived experience, research knowledge and other life skills and experiences, PRPs in both groups helped operationalize the research question, the project design and approach; conducted or participated in the literature review; collected data; and analyzed data or provided input in the analysis and interpretation of the results. During the initial stages of the project, the PRPs in the RLG influenced their group to conduct a literature review first before finalizing the design. The PRPs in the PLG influenced their group to conduct the formal review simultaneously with the first focus group. They used the information to develop their study materials as well as during data analysis. The PRPs in the RLG influenced their group to collect information from both clinicians and patients. The PLG collected information from only patients. The final list of attributes was reviewed and finalized with the PRPs in both groups. As such, the research and the list of attributes were relevant and reflective of the lived experience, beliefs and values of the PRPs in both groups. The stakeholders valued the experiences and knowledge that PRPs brought to the group. The resultant projects were patient-centered, collaborative, meaningful, rigorous, adaptable, ethical, legitimate, understandable, feasible, timely and sustainable.”